# Physicochemical Study of the Molecular Signal Transfer of Ultra-High Diluted Antibodies to Interferon-Gamma

**DOI:** 10.3390/ijms241511961

**Published:** 2023-07-26

**Authors:** Igor Jerman, Linda Ogrizek, Vesna Periček Krapež, Luka Jan

**Affiliations:** BION Institute, Stegne 21, 1000 Ljubljana, Slovenia; linda.ogrizek@bion.si (L.O.); vesna.pericek@bion.si (V.P.K.); luka.jan@bion.si (L.J.)

**Keywords:** ultra-high dilution, UHD signal transfer, molecular information, donor solution, receiver solution, physicochemical measurements, UV/VIS spectroscopy

## Abstract

Physicochemical investigations of (UHD) solutions subjected to certain physical factors (like shaking) are becoming more frequent and increasingly yielding convincing results. A much less studied phenomenon is the transfer of molecular information (UHD signals) from one fluid to another without an intermediate liquid phase. The purpose of this study was to investigate the possibility of such a UHD signal transfer from UHD solutions into the receiver fluid, especially when the molecular source used in solutions was a biologically active molecule of antibodies to interferon-gamma. We used physicochemical measurements and UV spectroscopy for this purpose. The results of this large pilot study confirm the possibility of such a transfer and a rough similarity to the original UHD signal donors, the weaker signal detection relative to the original donor fluids, and that exposure time improves the effect.

## 1. Introduction

From the standpoint of conventional chemistry and biochemistry, by dilution, any activity ascribed to substances in the original solution should diminish proportionally to the level of dilution. However, a growing body of published research is coming to quite different conclusions, considering aqueous solutions subjected to high dilution and special physical treatment. Sometimes, they go even beyond Avogadro’s limit and further (i.e., ultra-high dilutions 10^−12^ and more), where even theoretically, not a single molecule of the original substance(s) in aqueous solution should be left. These ultra-high dilutions will be denoted as UHD solutions. As to the physical treatment, most frequently, it represents mechanical vibrations, rotations, or electromagnetic field exposure during the dilution process or afterward [1]. The subject of the present research report concerns such processed aqueous solutions (liquids), which may be called processed liquids (solutions) that allegedly possess molecular imprints of the biologically active compounds which were initially present in physiologically effective concentrations. These imprints will be named UHD signals or molecular information and their further processing UHD signal transfer.

The research methods that demonstrate the UHD signal via differences between processed solutions to equally treated water with no diluted substances (i.e., control) are mostly physicochemical, physical [2,3,4,5], and even biological [6,7,8,9]. In numerous instances, processed solutions may not contain even a single molecule of the original substance(s) in the water. As a result, the effects observed cannot be attributed to chemistry but rather to the ultra-high dilution (UHD) signal that remains stored in the processed water, whether in liquid or solution form. While the precise nature of this signal remains elusive, it is generally assumed to reside within specifically ordered water molecular clusters or coherent domains forming the so-called mesoscopic water phase [10,11,12,13,14]. The most persuasive theoretical model is based on quantum electrodynamics (QED; [15,16,17]), although alternative explanations, such as Meesen’s nano-pearls [18] or clathrates [19], are also worth considering. Despite the inability of QED to provide exact mathematical predictions for experimental results in processing the ultra-high dilution (UHD) signal, it offers models that can serve as a comprehensive theoretical foundation for understanding molecular imprinting (i.e., the creation of the UHD signal) and the storage of this signal in a fluid medium through the aforementioned coherent domains of various types. The QED theory sees that as a process analogous to that in materials consisting of domains with magnetic moments [20,21].

In the scientific literature, various physicochemical and physical methods have been used to study UHD signal transmission with varying degrees of success. The present study used well-established methods, such as electrical conductivity (χ), pH, oxidation-reduction potential (ORP), and UV/VIS spectrometry. According to the established research practice, χ speaks about the concentration of dissolved ions in water. However, in studies of ultra-highly diluted solutions, a higher χ frequently indicates the presence of the above physical signal, i.e., molecular imprints in water [21,22,23,24]. It is frequently used in studying interfacial water bordering hydrophilic surfaces, where the hydronium ion (H_3_O^+^) may play the leading role [13,18] or dynamically ordered mesoscopic water states, also known as thixotropic phenomena [25,26,27] or structures such as nano-associates [28], clusters [29,30,31,32], domains [33,34,35], nanobubbles [36,37], nanoparticles [38], or naneons [39].

pH is frequently used in unconventional water research. Ryzhkina et al. found that with successive dilution and shaking, the pH in UHD solutions of 4-Aminopyridine decreases monotonically [24]; in other studies, they have found the pH to show non-linear response in UHD solutions compared to control [40], which was also confirmed in the systematic research report by Yablonskaya et al. [41]. These studies have indicated that, via pH measurements, we can detect more complex aspects of the UHD signals while they are processed through serial dilutions. In line with our previous research, we expected pH values to come closer to 7 when exposed to the UHD signal.

Although ORP measurements are infrequently used in non-conventional water research, it has been demonstrated that ultra-high dilutions combined with shaking result in raising the electrical voltage [42]. The separation of charges (rise of electrical tension) was also observed in specially structured water, also known as EZ water, adjacent to hydrophilic surfaces [43,44]. We, therefore, expected the UHD signal to increase the ORP at least slightly relative to the untreated (control) water.

As a review by Yinnon shows, UV/VIS spectrometry of serially diluted and vigorously shaken solutions is likewise a good marker of their ordered states [45]. Here, three different wavelength bands maxima are given: 205–210 nm, 260–280 nm, and 300–310 nm. The first quoted band is reported for the serially diluted and vigorously shaken solutions [24,40]. The second listed band is explored in EZ water research, where a broad peak absorption is regularly seen at 270 nm [46,47]. The third quoted absorption band (around 300 nm) is mostly studied in specially treated and presumably highly ordered MiliQ water [48]. In theoretical considerations, coherent domains, as postulated in QED, should be a major agent for a universally detected UV absorption of dynamically ordered UHD water solutions [45]. The relative absorption is expected to rise.

As already indicated, the UHD signal transfer research does not stop with dynamically ordered water structures. Surprisingly, studies by several teams have shown that the UHD signal not only changes the water in which an original substance has been diluted but can also be transferred to other liquids without direct contact between the donor and recipient solutions [49,50,51]. The nature of the UHD signal, therefore, appears to be complex and, at least at one of its levels, should go beyond dynamically ordered water structures; otherwise, it could not be transferred via glass, air, or magnetic field. Accordingly, we can assume that the signal in ordered water domains is a particular energy or field structure, which only in liquids uses mesoscopic structures (domains) for its residence. After certain physical conditions (such as shaking) are met, it can escape from its temporary abode and be taken up and stored by another, physically separated, liquid. Regarding the nature of the physical background of the UHD signal, most frequently, it is assumed to be electromagnetic in nature [51,52,53]. However, while it may exist in the form of coherent EM oscillations in mesoscopic water domains, out of the water, it may represent a quantum phenomenon or involvement of an as yet unknown quantum field or the field of quasi-particles; for instance, Kernbach proposes spin transfer [54]. In this case, roughly speaking, we should have a UHD signal in the donor solution resulting from dilution and shaking its transfer via a non-liquid medium (air, glass, EM field) and its reception in a fluid medium (receiver solution).

The above references clearly indicate that the UHD signal has been verified on multiple occasions. However, its specific physical nature has remained elusive for unequivocal confirmation. To better comprehend its nature and move closer to defining its physical properties, our team has conducted an extensive pilot study with three primary objectives:(a)To detect the underlying UHD signal in original solutions;(b)To identify the same signal in a further 100-fold dilution, along with simultaneous activation (C1 processing, see Section 4.1.2):(c)To detect the same UHD signal while attempting to transfer it from the donor solution to the recipient solution without an intermediate aqueous phase (physical transfer).

The overarching goal was:(1)To determine the key relations among the same kind of UHD signals in their three forms (original, C1 dilution, and physical transfer);(2)To find relations among the three chosen signals themselves, as they represent three distinct categories: a high and pure dilution of a biologically active compound, a mixed dilution of the latter, and just potentiated water.

For the biologically active compound that was subject to ultra-high dilution processes, we chose antibodies to IFNγ (marked as a-IFNγ) since the behavior of this specific active biological substance (and also a-IFNγ itself) has already been subject to extensive research with a noticeable outcome [1,55,56]. In the overall study, we used the a-IFNγ diluted in water with no further protecting measures, like dilution in alcohol or application to sugar pellets. Given some of the claims made by researchers [19] and our previous experience [57,58] that pure water is changeable and sensitive to new impressions coming from the surroundings, we did not expect the UHD signals to be very intense and clear, especially taking into account that the solutions have had to be transported across state borders. We used two different dilutions of a-IFNγ, namely a pure one (9 × 10^−2^ diluted), a mixed one with three different simultaneous dilutions (see Section 4.1.1 for more detail), and distilled water diluted in itself (the UHD signal of water; 9 × 10^−2^ “dilution”).

According to the goals and following our experiences and other research in the already cited literature, our assumptions were as follows:(1)We would successfully detect the UHD signal at all three processing stages by at least one measurement method;(2)The water (W) signal would probably be weaker than the pure antigen signal, which in turn would be weaker than the dilution mixture. This is due to the more complex processing of the dilution mixture, which allows for possible signal superimposition;(3)The physically transmitted signal would likely be weaker than either the original or C1-processed signal; however, a longer exposure time of the receiving liquid would result in its better detection;(4)The C1-processed signal might vary from the original signal in terms of intensity, reliability, and sign; however, in principle, it would follow the nature of the UHD signal from the original solutions.

## 2. Results

### 2.1. Original Solutions

Chemically, the original solutions were akin to distilled water, exhibiting low conductivity and slightly acidic pH. Despite common expectations of no statistically significant differences between substances, physicochemical measurements revealed significant differences in conductivity and pH between the a-IFNγ mix and all other liquids, as indicated by *p*-values and Cohen’s D. The a-IFNγ mix pH was found to be more acidic. As regards the ORP measurements, highly significant differences were observed between the WC9 solution and all other solutions (Wilcoxon test), accompanied by a very high Cohen’s D. Table 1 presents the calculated Cohen’s D values relative to the control (W). According to our expectations of the UHD signal working regarding different methods as presented in the Introduction (higher conductivity, higher ORP, pH approaching 7, and a higher UV relative absorption), the values are marked with different colors (green indicates a match to our expectations, red the opposite).

All three parameters had a non-normal distribution; therefore, we used the Friedman test. For both conductivity and pH, we found a statistically significant difference between the groups (conductivity: *p* = 0.037 and pH: *p* = 0.003). With ORP, we did not obtain a statistically significant difference between the groups; we only found a trend (*p* = 0.076). This is the reason for our performing a post hoc analysis with the Wilcoxon test for all three parameters. The results of the measurements are shown in Figure 1, Figure 2 and Figure 3 below.

Figure 1 shows the differences among samples in conductivity. We see that the a-IFNγ mix displays the highest and statistically significant difference in conductivity vs. all other samples. In parallel, we can observe large standard effect sizes (Cohen’s D, marked as d in the Figure).

Figure 2 shows the differences in ORP between the samples. The results show that the a-IFNγ mix has the lowest ORP value among all samples. It is unlike WC9, which has the highest and statistically significantly different ORP value vs. any other sample. In parallel, significant standard effect sizes (Cohen’s D, denoted as d in the Figure) can also be observed.

Figure 3 shows the differences in pH between the samples. It can be seen that, in line with our expectations written in the Introduction, the a-IFNγ mix has the lowest and statistically significantly different pH from any other sample. In parallel, high standard effect sizes (Cohen’s D, denoted as d in the Figure) can also be observed.

With UV/VIS spectroscopy of chemically almost pure water, we performed the measurements at 260 nm (Figure 4). The a-IFNγ mix shows the most noticeable and significant differences from the other three liquids (Figure 4a). It enhanced UV absorption vs. W; however, a-IFNγC9 showed reduced absorption, while W9 remained approximately at the same level.

### 2.2. Processing of Original Substance via Centesimal Dilution (C1)

After the C1 treatment, dilution of the original substances in RS1 resulted in similar conductivity measurements and slightly alkaline pH compared to the original solutions. Due to the non-normal distribution of conductivity (Figure 5), between-group differences were assessed using Friedman’s test (*p* = 0.000). Regarding ORP and pH, which exhibited normal distributions, ANOVA was used. No statistically significant differences were observed for either parameter. However, post hoc analysis showed significant differences between samples in ORP or pH measurements (Figure 6 and Figure 7), consistent with the trend observed in the measurements of the original solutions.

Figure 5 shows the differences between the samples in conductivity after the C1 processing. It can be seen that the a-IFNγ mix has the highest and statistically significantly different conductivity from all other samples. In parallel, we can also observe high standard effect sizes (Cohen’s D, denoted as d in the figure). The differences are very similar to the original substance (see Figure 1), but the effect seems to be slightly weaker.

Unlike the original substance, where the a-IFNγ had the lowest ORP (see Figure 2), here, it expresses the highest ORP. It is statistically significantly different from the control (W). However, the relationships are preserved, for example, between W:WC9 even after C1 processing (WC1:WC10) or between WC9:a-IFNγC9 after the same processing (WC10:a-IFNγC10) (see Figure 6).

Unlike in the original solutions, where the a-IFNγ mix had the lowest pH (see Figure 3), the a-IFNγC10 now has the lowest pH. However, the ratio of the WC9 to the a-IFNγC9 and the a-IFNγ mix remains roughly the same when we proceed (via C1 processing) to the WC10 vs. the a-IFNγC10 and the a-IFNγ mixC1 (see Figure 7).

With UV/VIS measurements, identical solutions were used for the C1 treatment and the physicochemical measurements, namely RS1. Preliminary tests have shown that a wavelength of 210 nm is the most suitable (for this spectral band, see Figure 8b). Here, we found three statistically significant differences (Figure 8a). The most notable is the difference between a-IFNγ mixC1 and a-IFNγC10.

Table 2 presents the calculated Cohen’s D values relative to the control (W). According to our expectations of the UHD signal working regarding different methods as presented in the Introduction (higher conductivity, higher ORP, pH approaching 7, and a higher UV relative absorption), the values are marked with different colors (green indicates a match to our expectations, red the opposite).

### 2.3. Physical Transfer

Immediately after activation, we did not observe any statistical difference in physicochemical measurements except in conductivity for the pair a-IFNγC9-sig. vs. control (W-sig.; Figure 9). ORP and pH measurements did not show any statistical difference.

After 1 h of exposure, the ORP of a-IFNγ mix-sig. showed a significant statistical difference against the control (W-sig.), which, as expected, increased with a longer exposure time—overnight (see Figure 10).

We did not detect any particular variation in the pH parameter, even during overnight exposure, where we observed substantial differences in conductivity and ORP (see Figure 11 and Figure 12). The exception is the exposure time of 1 h, where a significant statistical difference is seen, with a higher pH for the UHD signal.

Figure 11 below shows the difference between the W signal and the a-IFNγ mix signal in conductivity. It can be seen that the a-IFNγ mix signal yields a higher and statistically significantly different conductivity from the W signal. In parallel, we can also observe a high standard effect size (Cohen’s D, denoted as d in the figure). The difference is very similar to the differences observed from the original solutions (see Figure 1) and the differences after the C1 processing (see Figure 5).

Figure 12 below shows the difference between the a-IFNγ mix signal and control (signal W) in the ORP. We can observe that the a-IFNγ mix yields higher and statistically significantly different voltage vs. control. The differences are very similar to the differences found in the C1-processed solutions (as shown in Figure 5).

UV/VIS spectroscopy gave comparable results to the conductivity measurements, except that the exposure time was longer—1 h (see Figure 10).

Figure 13 below shows the differences in the relative UV/VIS absorption between the three tested signals compared to the control signal (W-sig.) after one hour of exposure. It can be seen from the Figure that, except for the WC9 signal, the absorption is higher than the control. However, it is more pronounced for the a-IFNγC9 signal, which is also statistically significant. It is similar to the results in conductivity immediately after activation (see Figure 9).

We found similarities between the physicochemical measurements and UV/VIS spectroscopy immediately after activation. They both yield statistically significant differences: in conductivity for the pair a-IFNγC9-sig.: W-sig., and with UV/VIS at the waveband of 190–200 nm with 1 h of exposure. We observed a similar trend between a-IFNγ mix-sig.: W-sig. and WC9-sig.: W-sig. with both measuring methods: conductivity and UV/VIS at the waveband of 190–200 nm. Table 3 presents the calculated Cohen’s D values relative to the control (W). According to our expectations of the UHD signal working regarding different methods as presented in the Introduction (higher conductivity, higher ORP, pH approaching 7, and a higher UV relative absorption), the values are marked with different colors (green indicates a match to our expectations, red the opposite).

### 2.4. Summary Comparisons Using Cohen’s D

Comparisons were also performed from multiple perspectives by calculating the average of the relative Cohen’s D values (compared to control: W, WC1, or W-sig.). This provided a rough estimate of the efficiency of the four measurement methods. Figure 14 compares the performance of these methods concerning the three original solutions, C1-treated solutions, and physical transfer of the UHD signal. For all three treated substances, conductivity exhibited the highest measurement efficiency, followed by UV/VIS and ORP measurements, while pH measurements demonstrated the lowest performance (here, the negative Cohen’s D supports signal working expectations). A similar result was observed for C1 processing through all detection methods.

Summarizing the results of all the methods used, Figure 15 below shows the overall performance of the substances. It may be seen that the strongest original signal is observed in the a-IFNγ mix. The C1 processing seems to make the a-IFNγC9 signal more apparent, which is not observed for WC9, which remains approximately at the same level (but still higher than a-IFNγC9), in contrast to the a-IFNγ mix, where the signal becomes weaker. For the UHD transfer, the most significant effects are observed with a-IFNγC9, while the smallest differences are seen with WC9. It is worth noting that the a-IFNγ mix shows comparable signal effect sizes for the original solution, C1 dilution, and physical transfer.

Comparing all three substances across all experiments, we note that the signal of WC9 is weaker than the a-IFNγ mix or a-IFNγC9. There is also a noticeable difference between the a-IFNγ mix and the a-IFNγC9 signals, where it can be seen that overall, the stronger signal is from the a-IFNγ mix, as shown in the pie chart below (Figure 16).

## 3. Discussion

### 3.1. Possibility of UHD Signal Detection

In general, the present study confirms the existence of the UHD signal, thereby corroborating many similar scientific research reports. As already discussed in the Introduction, the conventional physical and chemical explanations do not provide meaningful answers. However, quantum electrodynamics (QED) may serve as a valuable explanatory background. The QED applied to water and quantum vacuum oscillations discovered the emergence of spontaneously ordered (coherent) domains in water. Having low entropy, no energy is required for the maintenance of order [59]. In iterated dilutions, characteristic for UHD, the detected signal is most probably due to this property of coherent domains that overcome the information destruction by overcoming the thermal noise, kT (see more in [60]). The process of repeated dilution is necessary to minimize the noise created by dissolved ions in a water-based solution, which are constantly moving due to Brownian motion. This dilution reduces the concentration of selected ions that are capable of causing the observed effect after dilutions. However, the signal we anticipate from these ions, based on the Liboff–Zhadin effect, is amplified, effectively counterbalancing the dilution through the protonation of water. This amplification is attributed to Schumann frequencies, the highest peak of which coincides with the ion cyclotron frequency of any hydronium ion hydrate, particularly in temperate latitudes [61]. Thus, thanks to the noise suppression operated by dilution, the energy gap provided by pure water coherent domains [16] is sufficient to overcome the kT Brownian noise, thus allowing the maintenance of the UHD signal.

### 3.2. Hypotheses

Our extensive, albeit pilot, present study of UHD signals obtained by further dilution (C1) and physical transfer, and measured by four different methods, confirmed the detectability of the signals as proposed in our first hypothesis. Generally speaking, the results concerning the simultaneously used four detection methods conform to our expectations (see, for instance, Figure 14). The detection of the WC9 or WC10 signals can be attributed to the shaking itself [62,63], and as seen in Figure 15, the results follow our second hypothesis that the water (W) signal would probably be weaker than the antigen signals. Another expectation expressed in the second hypothesis, namely that the pure antigen signal would generally be fainter than the signal detected from the dilution mixture, has also been corroborated (see Figure 15).

Our third hypothesis assumes that the signal transferred via glass will be detectable by at least one of the four methods utilized and that its strength will be fainter than that of the original solutions. The results confirmed the assumption; furthermore, the physical transfer of the UHD signal was detected by all methods used (Figure 14). Conductivity exhibited the most significant differences, followed by UV/VIS. As proposed, the transmitted signals proved weaker than the ones from the original solutions. We may observe the similarity of the transferred signal to the original if we compare the d-values between the original samples and those after the UHD transfer (for example, see the color correspondence between Table 1 and Table 3). The exposure time experiments also confirmed that longer exposure (even for one hour) resulted in more distinct differences (a-IFNγ-sig. vs. W-sig. (control) and a-IFNγ mix-sig. vs. W-sig. (control) in ORP (Figure 10 and Figure 12) and conductivity (Figure 11).

In the fourth hypothesis, we proposed that the C1-processed signal would, in principle, follow the UHD signal from the original solutions. The comparison of Table 1 and Table 2 (see also the color coding) supports this general assumption (six strong matchings vs. two strong non-matchings). We observe the best match with the a-IFNγ mix.

### 3.3. Research Methods

Following the published research and our own previous experiments, we assumed that the signal would be most detectable by measuring conductivity. As can be seen in Figure 14, this is true for all three signal types combined, followed closely by UV/VIS. Our tentative explanation here is that the UHD signal—including that produced by the water itself—increases dynamic orderliness in water, as already addressed in the Introduction (thixotropic phenomena [25,26,27], Konovalov and Ryzhkina’s review article [28]). As largely discussed in the article concerning the relationship between raised conductivity and dynamic order in water, the correlation of proton hopping over long hydrogen-bonded water chains may arise [64]. Thus, Lapid and colleagues [65] propose that proton mobility is characterized by a cooperative phenomenon, resulting in facilitated proton conduction when water molecules are more ordered [66,67]. The higher degree of orderliness of water is also confirmed through theoretical estimates by various authors based on quantum electrodynamic theory (QED). They attribute the working of the UHD signal to coherent domains (i.e., stable domains of dynamically ordered water), introduced in the Introduction (see [15,16,17,20,21]), and may also be found in the article by Elia et al. [68].

Regarding ORP, we expected the UHD signal would increase the electrical voltage (and hence the ORP). In the original substances (Figure 2), the ORP was lower for both UHD signals of the biologically active compounds vs. WC9. The C1 processed signal, however, corroborated the expectations (Figure 6 and Figure 14), as was also the case with the physical transfer of the UHD signal (Figure 12). As already discussed in the Introduction, this may be due to vigorous shaking and the production of coherent domains, as discussed more thoroughly in [21,42], involving QED theory.

Regarding pH, we assumed that the UHD signal would support the dissociation of ions, thus lowering pH in acidic solutions and raising pH in alkaline ones. The results of the study show the following: The pH of UHD signals of the original solutions showed a decrease except for WC9, which confirms the assumption since, in this case, the measured liquid represented almost distilled water with a slightly acidic nature due to the dissolved CO_2_. The signal seems to enhance the dissociation of H_2_CO_3_ into H_3_O^+^ and HCO_3_^−^. As the concentration of HCO_3_^−^ increases, so does the H_3_O^+^ concentration, resulting in higher acidity. In other experiments, while using RS1 with the dissolved NaHCO_3_, the liquids were slightly alkaline due to the Na^+^ cation. An enhanced dissociation should, therefore, yield a higher OH^−^ concentration, consequently raising the pH. The apparent contradictory effect, where the UHD signal increases acidity in some cases and alkalinity in others, manifests itself as one impact: facilitating ion dissociation. The C1-processed solution did not conform to this expectation (lower or equal pH of UHD signals, see Figure 7; and in the physical transfer of the signal after one hour of exposure (Table 3, see pH result for the a-IFNγ mix-sig, column “10”).

As to the UV/VIS absorption, based on similar published research [40], we expected that the UHD signal absorption would enhance the UV absorption. Overall, this assumption was confirmed since, in general, in all cases of UV/VIS spectroscopy, the original (Figure 4a, Table 1), C1-processed signals (Figure 8a,b, Table 2), and physically transferred UHD signals of biologically active compounds (Figure 13, Table 3) demonstrate a higher absorption than the control regardless of whether or not it is statistically significant. As to the physical UHD signal, it is of note that Dibble et al. also found quite a similar effect applying a signal denoted as “subtle energy input” to water [69]. A higher UV absorption rate of liquids may be due to the UHD signal captured to some extent by the coherent domains of water molecules. In the latter, a larger fraction then transits between ground and excited electronic states, as predicted by QED [45].

The pilot study supports most assumptions from previous research and raises new questions that warrant further systematic studies. These include exploring the variability of fluid samples over time, the effects of sequential processing, and the stability of ultra-high dilution (UHD) signals. The most intriguing aspect is the physical transfer of UHD signals, which could be explained by quantum electrodynamic theory. However, understanding the transfer through glass and air remains unclear. Some experiments suggest that environmental electromagnetic or magnetic fields may be involved in the transfer process. Further research should focus on conducting experiments in various electromagnetic and magnetic environments to develop a theoretical model for this phenomenon, as supported by studies from Montagnier et al. [70] and Tang et al. [53].

## 4. Materials and Methods

To minimize potential experimental errors, we consistently implemented the same procedure for all samples, both control and treated, within the same experimental setup.

In the scientific water research area of not yet generally accepted and theoretically well-established phenomena, research using various detecting and measuring methods abound; however, there is not yet generally accepted methodology. Moreover, there is no specific sensor for dynamically ordered states of mesoscopic water. In this case, we need to use a statistically well-founded measurement system. To ensure maximal reliability of measurements, the BION Institute water research team devoted a special effort to developing and calibrating physicochemical and UV/VIS spectroscopical measuring protocols. They were developed and tested in order to be simultaneously robust and sensitive enough to detect subtle investigated phenomena. The system developed so far and used in the present study includes physicochemical methods (pH, ORP, and conductivity) and UV-VIS spectrometry.

### 4.1. Liquids

#### 4.1.1. Original (Received) Solutions and Waters

All original substances (water and solutions, four different liquids altogether) subject to research were obtained from OOO “NPF ”Materia Medica Holding”, Moscow, Russia. They were the unprocessed (not diluted or shaken) distilled water that played the role of control in all experiments. It was the basis for other dilutions and is marked as W. Another liquid was the same water processed as if diluted in itself by nine centesimal sequential dilutions (9 × 10^−2^). It was consecutively shaken between each “dilution” and is marked as WC9. The third sent liquid, prepared by Materia Medica, was a processed dilution of antibodies to interferon-gamma (a-IFNγ). It was processed in the same nine-fold centesimal manner as the second liquid (9 × 10^−2^) and is marked as a-IFNγC9. The fourth liquid is a multiple-processed dilution of a-IFNγ. The liquid represents the following mixture of successive centesimal dilutions (C) with intermediate shaking: C12, C30, and C50 (a-IFNγ C12, C30, C50) and is marked as the a-IFNγ mix.

The samples were supplied in polypropylene flasks, but before the experiment, they were poured from polypropylene into glass flasks (Duran^®^ GL 45, UAB Santonika, Kaunas, Lithuania), protected from intense direct light, and stored at room temperature with closed lids (20–23 °C).

#### 4.1.2. Further Centesimal (C1) Processing of Solution

In order to study the nature of the UHD signal after further dilution and activation, the original solutions were diluted 100 times (10^−2^) in one step, followed by 100 blows on the glass of a Duran^®^ bottle with a wooden mallet (C1 treatment already defined). The estimated force exerted during the manual strokes is approximately 0.04 N, with a frequency of 2 Hz. In this case, the original solutions were not diluted in distilled water but in a special solution with the following composition, marked as RS1: 0.413 mL of 3% hydrogen peroxide (Lekarna Ljubljana Pharmacy, Ljubljana, Slovenia) per liter (12.39 mg/L of pure H_2_O_2_; i.e., 0.36 mM H_2_O_2_) and 0.2 g/L sodium hydrogen carbonate (Solvay Chimica Italia S.p.A, Rosignano, Italy; 0.01 M NaHCO_3_) diluted on distilled water provided by the Bion distiller (χ = 2 µS). Therefore, the original distilled water (W) became WC1, WC9 became WC10, a-IFNγC9 became a-IFNγC10, and the a-IFNγ mix became a-IFNγ mixC1.

#### 4.1.3. Receiver Solutions

In the study of further diluting of the received substances as defined in Section 4.1.1, we used various receiver solutions for investigating the contact transfer of molecular information from the donor (original) substances. Following our previous experiences and additional examination of optimizing possible liquids for the present investigation, as well as taking into consideration the profound study by Voeikov [71], we used the already mentioned and defined RS1 containing traces of hydrogen peroxide and hydrogen carbonate.

### 4.2. Physical Transfer of UHD Signal via Glass

#### 4.2.1. General Methodology and Distance

Principally, we performed a physical transfer of the UHD signal of the original substances (donor) via glass or air by applying strokes to the bottle with the donor liquid in the direction of the bottle with the receiver solution. This transfer method is called “with activation”; it means that the glass bottle with the donor substance is shaken 15 times with a mallet shaker while the vibrations are transmitted from the first bottle to the second bottle. The transfer with activation was performed via glass, where the donor bottle was touching the receiver bottle. While doing the air transfer through shaking, the donor bottle had no contact with the surface on which the bottle of the receiver was placed. We mark the transferred signals from the original substances as W-sig. (from W), WC9-sig. (from WC9), a-IFNγC9-sig. (from a-IFNγC9) and a-IFNγ mix-sig. (from the a-IFNγ mix).

#### 4.2.2. Exposure and Separation Times

We also studied different times of exposure of the receiver solution to the UHD signal. In most cases, the measurements were taken immediately after activation. In addition, we also studied the reception and expression of the UHD signal after different exposure times and a one-hour separation of the donor and receiver bottles by moving the previously attached bottles away from each other. The purpose of this experiment was to repeat the measurement of the solutions’ parameters after one hour of separation of the bottles. These different time experiments will be presented in detail in another publication; here, we will disclose the following experiments:(a)immediately after activation (mark: 00);(b)one hour of exposure after activation (mark: 10);(c)overnight exposure (mark: N0).

### 4.3. Measurement Methods

#### 4.3.1. General

For measurement of the physicochemical parameters, we used The Vernier Go Direct^®^ devices (Vernier Software & Technology, Beaverton, Oregon, United States): Temperature Probe, pH Sensor, Conductivity Probe, and ORP Sensor. We employed simultaneous temperature measurements as a control to ensure that the observed data differences did not result from temperature variations. Conductivity is known to have a strong correlation with temperature, and to a lesser extent, this also pertains to the other measurement methods used. Each individual measurement was conducted in a separate beaker. The measured range of accuracy (which deviates from the officially declared value and is not part of the observed, investigated, and considered measurement drift) for these three devices is as follows: 0.5 µS/cm for conductivity, 2 mV for ORP, and 0.02 for pH.

For UV/VIS absorption spectroscopy measurements, we used a Macherey–Nagel spectrophotometer, wavelength range 190–1100 nm, with a 50 mm quartz cuvette cell. The measured accuracy range here is 0.003. Here, too, each measurement was performed in a single glass beaker.

#### 4.3.2. Measurement Protocol

For the physicochemical measurements of the original substances, the liquids obtained in the bottles were poured into beakers. All physicochemical parameters (conductivity, ORP, pH) and temperature were measured simultaneously by two sets of measuring devices. In order to prevent possible slight measurement errors due to the drift of measurements over time, we did not measure different solutions consecutively but alternately. Regarding the receiver water, the RS1 was found to be suitable for testing further processed solutions (C1 experiments, physicochemical method). Physicochemical methods such as conductivity and pH have also been used in various other studies to detect signals in ultra-high diluted solutions [41,72,73].

For UV/VIS absorption measurements of the original substances, the liquid was poured into a 50 mm quartz cuvette cell, wherein measurements were performed alternately with different solutions. All the measurements were relative to the reference (W). Due to possible slight measurement drift, we performed the measurements according to the procedure explained for the physicochemical measurements above. According to some other authors, the strongest signal is detected at 260 nm in the original UV/VIS spectroscopy measurements [24,40,43,74]. Furthermore, we also obtained a statistical difference between the substances and the control at 260 nm.

For UV/VIS measurements of C1-processed solutions, initially, we used the previously explained dilution methodology (see Section 4.1.2) and then poured the newly produced liquids into a 50 mm quartz cuvette cell that we used for comparison. Again, all measurements were relative to the reference W (centesimally (10^−2^) diluted in RS1. We decided to compare the substances at 210 nm wavelength.

For analyses, we used the wavelength of 260 nm with the original substances and 210 nm with C1-processed samples due to a different liquid composition. Due to a larger variation in the UV absorption with the physical transfer of the UHD signal, we decided not to choose a particular wavelength but the average absorption values of the band spanning from 190–210 nm.

### 4.4. Experimental Procedures

The following Table 4, Table 5 and Table 6 present the experimental procedures according to the number of individual measurements per substance and the method used: electrical conductivity (χ), pH, ORP, and UV/VIS. All tested situations in all experiments had the same treatment and exposure to (un)controlled environmental factors. Measurement drift and temperature were taken into account as well.

The cumulative number of replicate measurements made with the original substances and the three physicochemical methods is shown in Table 4.

Table 5 below shows the number of replicate measurements performed for the C1-processed substances comprising all three physicochemical methods.

Table 6 shows the number of repeated UV/VIS measurements taken with both the original and C1 solutions. Due to the high sensitivity of the UV/VIS device, a lower external noise (from the measurements procedure) compared to physicochemical methods, and a better absorption rate in RS1, we decided to conduct fewer measurements for C1-processed liquids.

### 4.5. Experimental Plot

Measurements were performed with all four substances (W, WC9, a-IFNγC9, and a-IFNγ mix) using physicochemical methods and UV/VIS spectroscopy. In experiments applying further C1 processing, the liquids were subjected to the treatment described in Section 4.1.2 (see Table 7, second experimental group). We compared the results of all measured solutions with each other.

In the third experimental group (Table 7), we performed various experiments on the physical transfer of the UHD signal. We compared the RS1 solution of the transferred control signal (W) with the RS1 solutions of other transferred substances at different distances: 0, 1, and 4 cm. However, in the present paper, we will report only about the experiments via glass (distance 0) that were conducted with physicochemical methods immediately after activation (marked as 00, all four substances), after one hour of exposure (marked as 10), and after overnight exposure (marked as N0).

### 4.6. Statistical Analysis of the Results

For estimating statistical significance, we applied appropriate tests regarding normality and the number of groups. In the case of the normal distribution, we used ANOVA; otherwise, we used the Friedman test. For post hoc analysis, we subjected the groups to a post hoc *t*-test (normal distribution) or/and Wilcoxon signed-rank test. All tests were performed via pairwise comparison.

Statistical data analysis was performed using XLSTAT statistical software (XLSTAT PREMIUM-Evaluation 2022.3.1) for Excel. For basic statistical parameters of groups, we calculated the average, standard deviation, standard error, and normality with the Shapiro–Wilcoxon test. We consider that the pairwise method of the Shapiro–Wilcoxon test is appropriate for two reasons: (a) the samples are not unrelated, as they have the same original distilled water, (b) there is often a slight drift in the measurements, which can lead to a systematic measurement error in the required precision range in detecting statistically significant differences. To estimate statistical significance in the data variation, we used Levene’s test or F-test for differences in variance and Cohen’s D for assessing standardized effect size.

Differences are considered statistically significant with *p* < 0.05 or if 0.1 < *p* < 0.05, while the absolute Cohen’s D value is higher than 0.5.

To make various comparisons, not only between the substances in their original or processed forms but also between the efficacy of the used physicochemical methods and the methods of UHD signal treatment, we resorted to the calculation of the average of absolute Cohen’s D values, where the latter were calculated vs. control (W). Thus, we could have made synthetic evaluations of the study results from different perspectives.

## 5. Conclusions

This extended pilot study confirmed the measurable presence of UHD signals stored in water. The phenomenon was not observed only in liquids where originally a biologically active substance was diluted but also in physically processed distilled water (WC9), although the detection here was less reliable.

C1 processing yielded comparable results to the original solutions, although two of them (of twelve altogether) went in the opposite direction. The best score of matching was achieved by the a-IFNγ mix signal.

After the physical transfer via glass, the signal’s detection was weaker than the one from the original samples. The results may have diverged from the original, although it is difficult to compare the two since the experimental setups (different times of exposure, the selection of original substances) do not quite fit together.

Regarding the methods, the most effective proved to be conductivity measurements, followed closely by UV/VIS.

Summarizing the measuring methods and the signal processing, the a-IFNγ mix showed the highest difference to the control (unprocessed water, W), the a-IFNγC9 was slightly lower; as expected, the processed water (WC9) presented the lowest overall difference to the control.

## Figures and Tables

**Figure 1 ijms-24-11961-f001:**
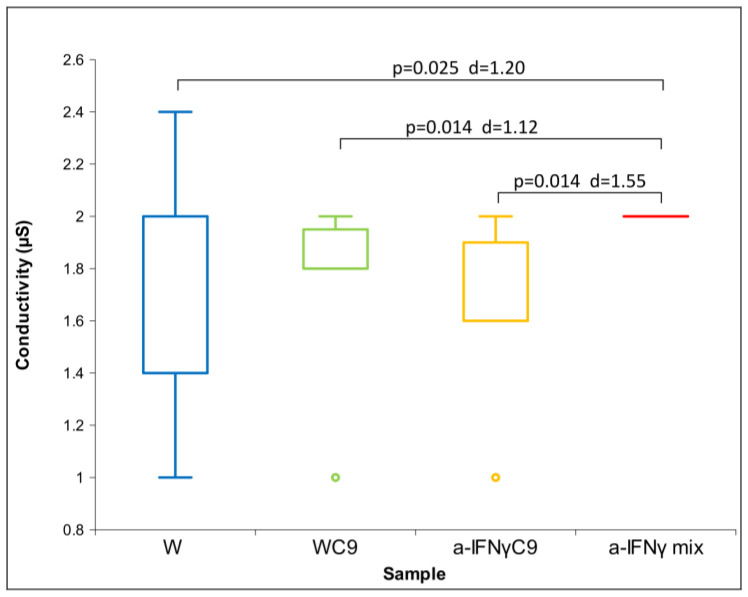
Box plot of conductivity measurements presenting median and quartiles of original solutions (N = 10). Significances (*p*-values and d-values) in differences are presented, too.

**Figure 2 ijms-24-11961-f002:**
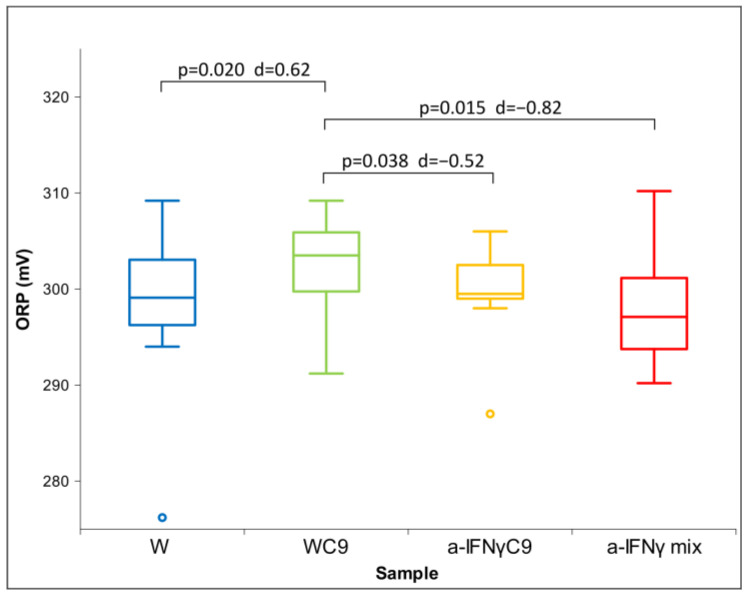
Box plot of ORP measurements presenting median and quartiles of original solutions (N = 10). Significances (*p*-values and d-values) in differences are presented, too.

**Figure 3 ijms-24-11961-f003:**
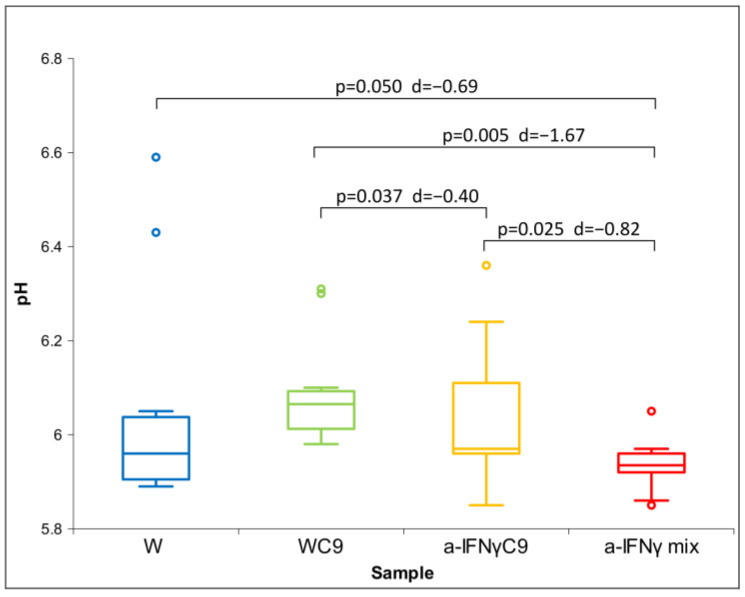
Box plot of pH measurements presenting median and quartiles of original solutions (N = 10). Significances (*p*-values and d-values) in differences are presented, too.

**Figure 4 ijms-24-11961-f004:**
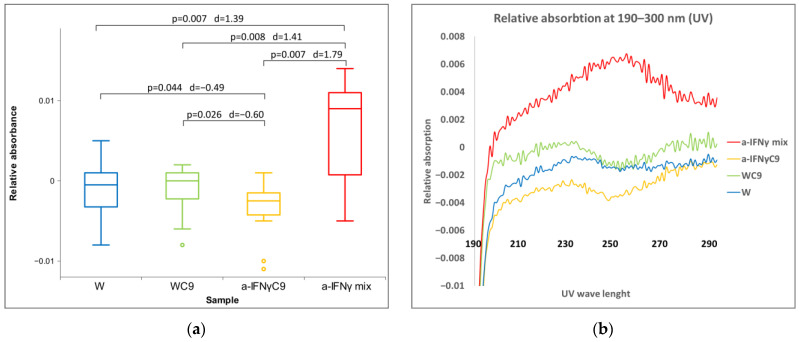
(**a**) Box plot presenting median and quartiles of original solutions of relative absorbance at 260 nm (N = 12). Significances (*p*-values and d-values) in differences are also presented; (**b**) relative absorbance of means: original, at 190–300 nm (N = 12).

**Figure 5 ijms-24-11961-f005:**
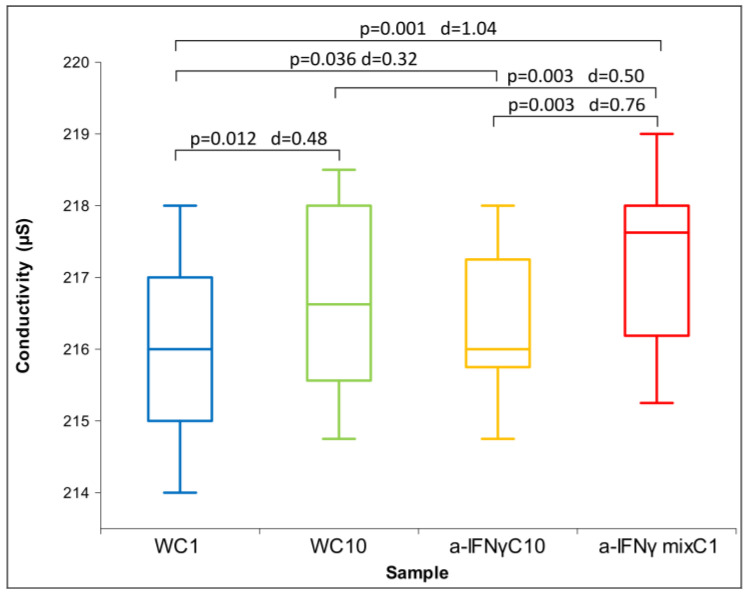
Box plot of conductivity measurements presenting median and quartiles of C1 (N = 16). Significances (*p*-values and d-values) in differences are also presented.

**Figure 6 ijms-24-11961-f006:**
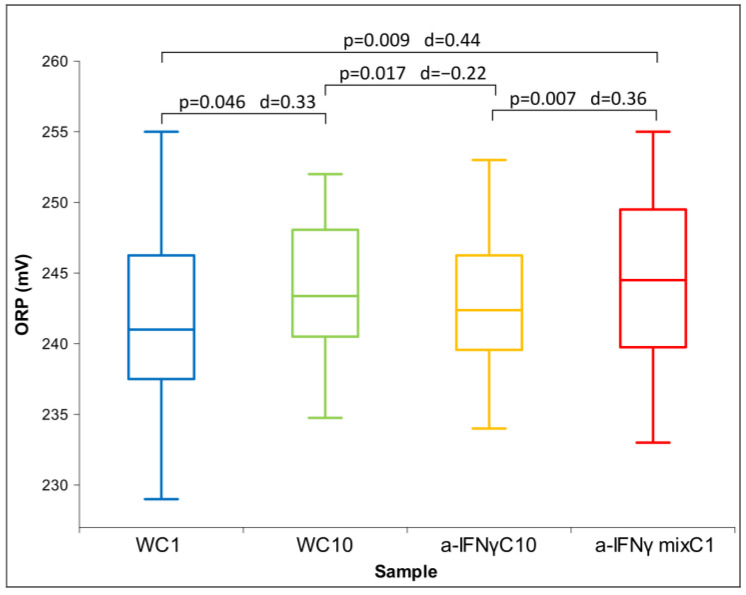
Box plot of ORP measurements presenting median and quartiles of C1 (N = 16). Significances (*p*-values and d-values) in differences are also presented.

**Figure 7 ijms-24-11961-f007:**
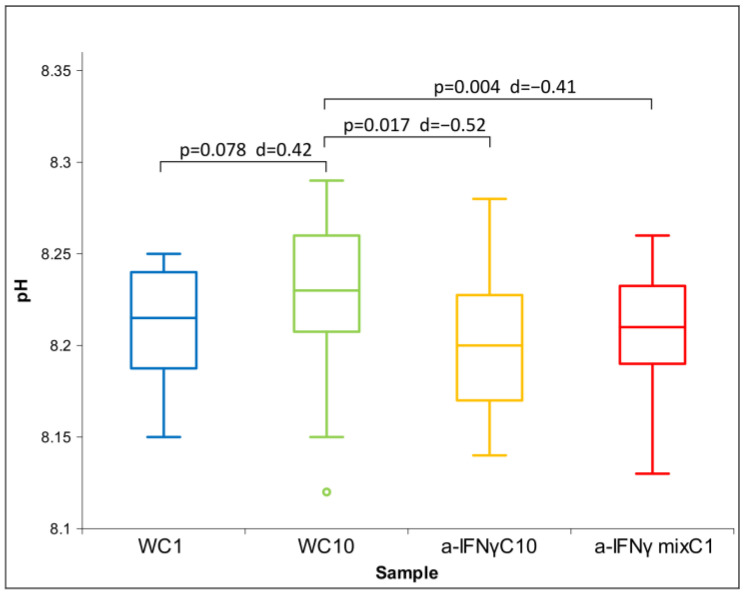
Box plot of pH measurements presenting median and quartiles of C1 (N = 16). Significances (*p*-value or *p*-value with d) in differences are also presented.

**Figure 8 ijms-24-11961-f008:**
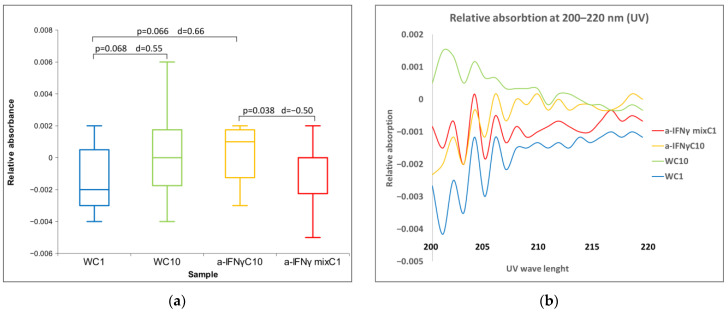
(**a**) Box plot presenting median and quartiles of C1, of relative absorbance at 210 (N = 12). Significances (*p*-value or *p*-value with d) in differences are also presented; (**b**) relative absorbance of means: C1, at 200–220 nm (N = 12).

**Figure 9 ijms-24-11961-f009:**
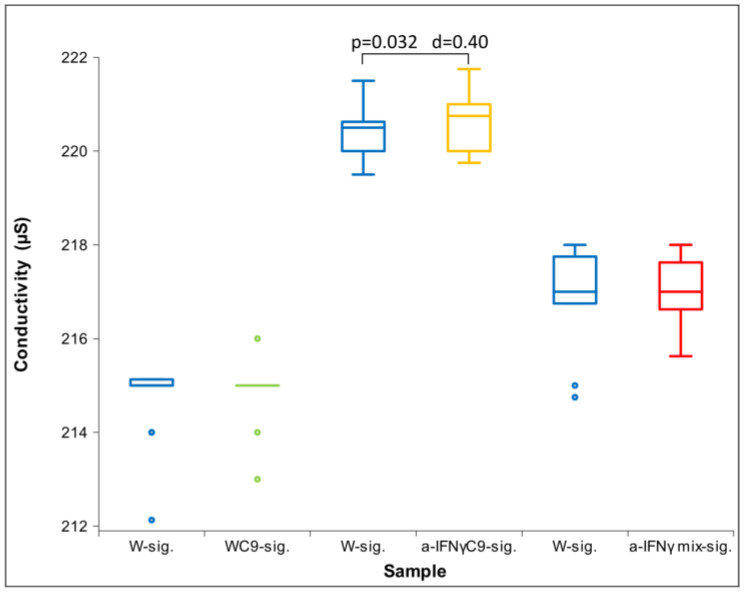
Box plot of conductivity measurements presenting median and quartiles, transfer via glass immediately after shaking (N = 16). Significances (*p*-values and d-values) in differences are also presented.

**Figure 10 ijms-24-11961-f010:**
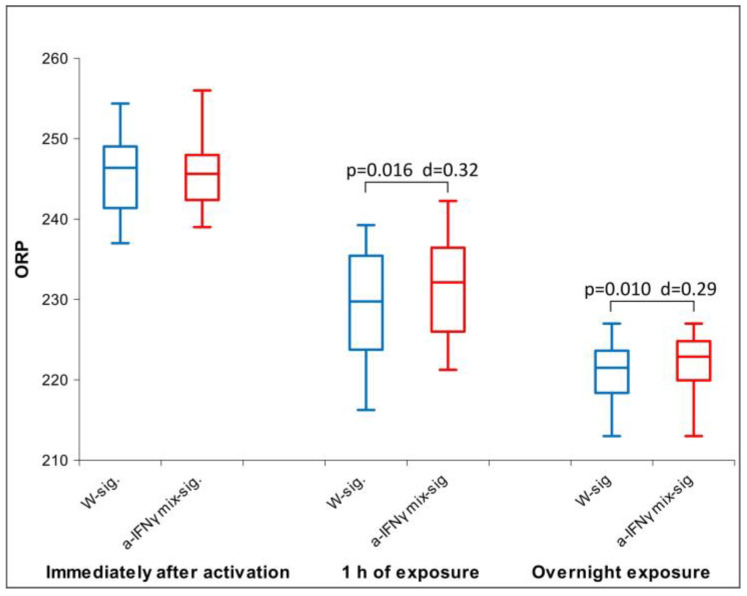
Box plot of ORP measurements presenting median and quartiles of ORP with different times of exposure (N = 16). Significances (*p*-values and d-values) in differences are also presented.

**Figure 11 ijms-24-11961-f011:**
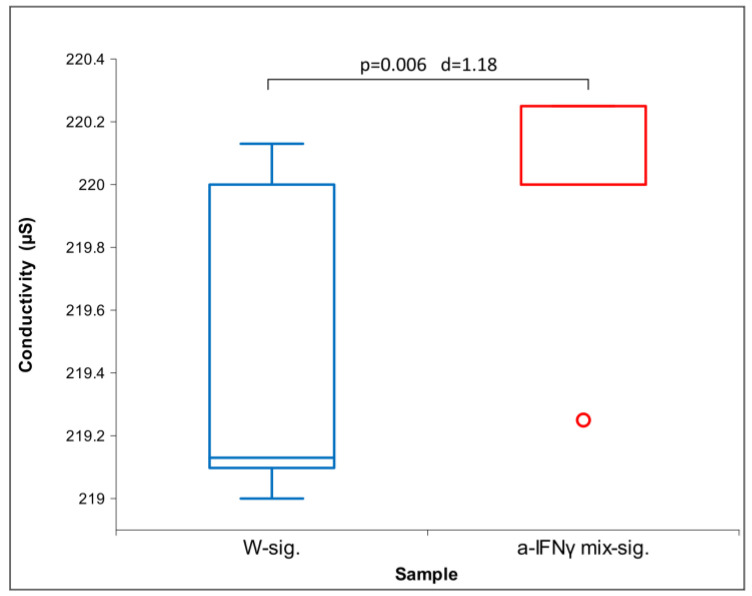
Box plot of conductivity measurements with data means and standard deviation: transfer via glass, overnight exposure (N = 16). Significances (*p*-values and d-values) in differences are also presented.

**Figure 12 ijms-24-11961-f012:**
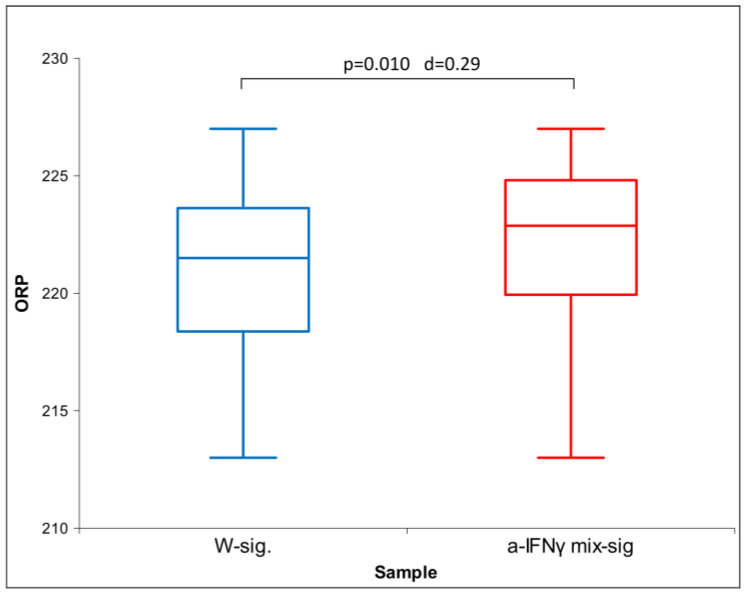
Box plot of ORP measurements with data means and standard deviation: transfer via glass, overnight exposure (N = 16). Significances (*p*-values and d-values) in differences are also presented.

**Figure 13 ijms-24-11961-f013:**
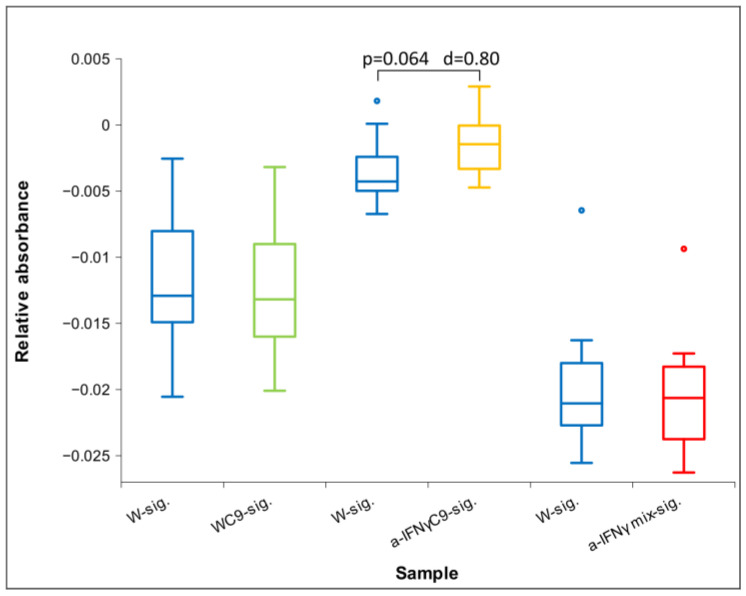
Box plot of relative absorbance at the waveband 190–200 nm for transfer after 1 h exposure (N = 12). Significance (*p*-value with d) in difference is also presented.

**Figure 14 ijms-24-11961-f014:**
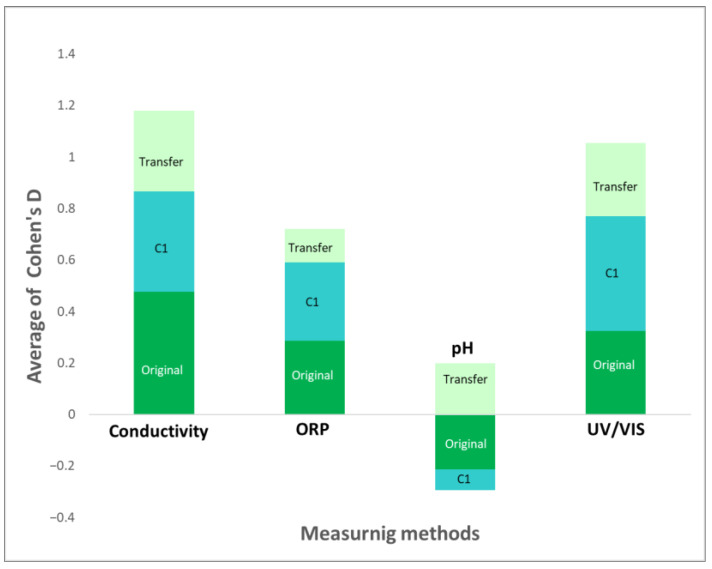
Comparisons regarding the effectiveness (in average Cohen’s D) of the four measuring methods involving the original substances, C1 processed solutions, and the physical UHD signal transfer.

**Figure 15 ijms-24-11961-f015:**
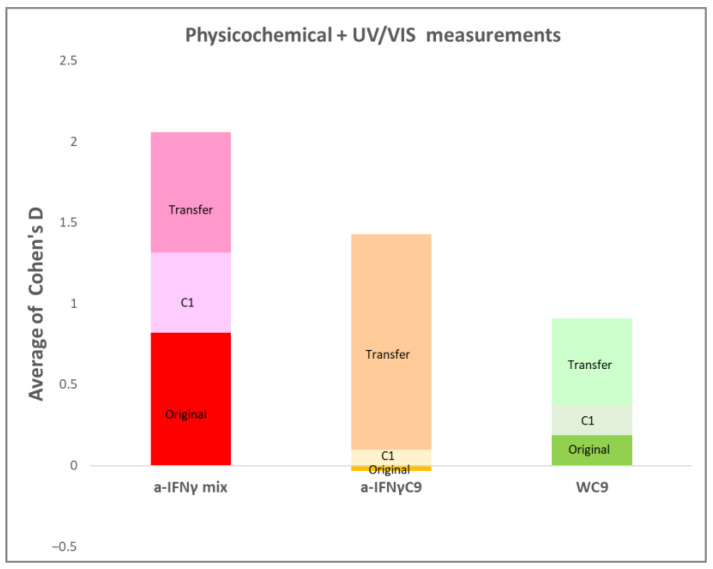
Cohen’s D relative mean values comprise all measurement methods for all three kinds of UHD signal: original, C1-processed, and physically transferred.

**Figure 16 ijms-24-11961-f016:**
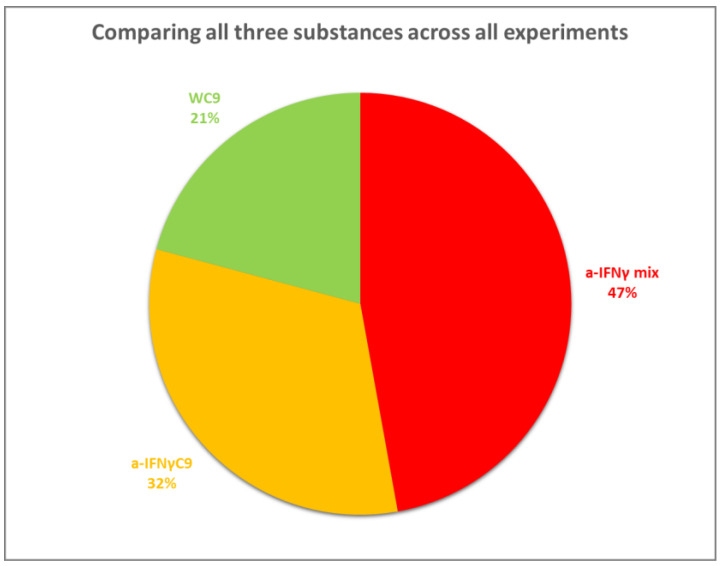
The summary pie chart of Cohen’s D relative mean values comprising all four methods for all three kinds of UHD signals.

**Table 1 ijms-24-11961-t001:** Standardized effect sizes of the original samples compared to water (control) for three different physicochemical measurement methods and UV/VIS spectroscopy. Values that are statistically significant (*p* < 0.05) are in bold. The values supporting our expectations are highlighted in green, and those opposing them are in red.

Sample	Conductivity	ORP	pH	UV/VIS
a-IFNγ mix	**1.20**	0	**−0.69**	**1.39**
a-IFNγC9	0	0.24	−0.12	**−0.49**
WC9	0.23	**0.62**	0.17	0.07

**Table 2 ijms-24-11961-t002:** Standardized effect sizes of C1-processed original samples physicochemical and UV against water (control). Values that are statistically significant (*p* < 0.05) are in bold. The values that support our expectations are highlighted in green, and those that do not are in red.

C1-Processed Samples	Conductivity	ORP	pH	UV/VIS
a-IFNγ mixC1	**1.04**	**0.44**	−0.02	0.13
a-IFNγC10	**0.32**	0.14	−0.16	**0.66**
WC10	−0.19	**0.33**	**0.42**	**0.55**

**Table 3 ijms-24-11961-t003:** Standardized effect sizes of physical transfer of original samples physicochemical and UV against water (control). Marks: 00—measurements immediately after activation (shaking), 10—measurements after one hour of exposure after activation, N0—measurements after overnight exposure after activation. Values that are statistically significant (*p* < 0.05) are in bold. The values supporting our expectations are highlighted in green, and those opposing are red marked.

	Conductivity			ORP			pH			UV/VIS		
Sample/Time	00	10	N0	00	10	N0	00	10	N0	00	10	N0
a-IFNγ mix-sig.	0	−0.18	** 1.18 **	0	** 0.32 **	** 0.29 **	0.05	** 0.62 **	0.25	** 0.27 **	0.14	0.21
a-IFNγC9-sig.	** 0.40 **	/	/	−0.05	/	/	0.23	/	/	0.09	** 1.08 **	/
WC9-sig.	0.17	/	/	0.09	/	/	−0.15	/	/	−0.07	0.27	/

**Table 4 ijms-24-11961-t004:** The number of repeated measurements of the original liquids for the three physicochemical measurement methods.

Substances	χ	ORP	pH
a-IFNγ mix	10	10	10
a-IFNγC9	10	10	10
WC9	10	10	10
W	10	10	10

**Table 5 ijms-24-11961-t005:** The procedure of physicochemical measurements of the C1 processed original solutions, number of measurements.

Substances	χ	ORP	pH
a-IFNγ mixC1	16	16	16
a-IFNγC10	16	16	16
WC10	16	16	16
WC1	16	16	16

**Table 6 ijms-24-11961-t006:** The procedure of UV/VIS measurements of the original substances and C1-processed liquids. W was measured simultaneously and used as a reference.

Substances	UV/VIS Original	Substances	UV/VIS C1 (RS1)
a-IFNγ mix	12	a-IFNγ mixC1	6
a-IFNγC9	12	a-IFNγC10	6
WC9	12	WC10	6
W	12	WC1	6

**Table 7 ijms-24-11961-t007:** For all three experimental groups, we used all four methods (three physicochemical and UV/VIS). In the first two groups, we performed the same number of experiments. In the third group, where the experimental situations were more wide-ranging, the number varied.

Experimental Groups	Methods and Techniques	Number of Experimental Situations
1. Original substances	Physicochemical methods	1
	UV/VIS spectroscopy	1
2. C1 processed original substances	Physicochemical methods	1
	UV/VIS spectroscopy	1
3. Physical UHD signal transfer	Physicochemical	5
	UV/VIS	7

## Data Availability

Research data are available from the BION Institute upon special request.

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
