# Peer review of "Physicochemical Study of the Molecular Signal Transfer of Ultra-High Diluted Antibodies to Interferon-Gamma"

_ijms, 2023, doi:10.3390/ijms241511961_

Round 1

Reviewer 1 Report (Previous Reviewer 2)

The article is more structured than the original material. The authors have done a great job. However, there are still remarks that need to be corrected:

1. Introduction
- I recommend to the authors to change the purpose: As the comprehension of the UHD signal's nature currently presents a challenge to established science, the goal of this study was to confirm its presence in UHD solutions and also, to investigate the transfer of the UHD signal from one solution to another without a liquid bridge.

2. Figures
- Figures 4b,8b,14,15: it is necessary to remove the grid, as well as to change the color of the frame, i.e., instead of light grey, make it dark grey frame similar to the other Figures. 

3. Tables
- Tables 1-3: I recommend to the authors to remove the color coding of Tables. It is not clear why it was introduced. If this coding is fundamental, then below the Tables need to give a deciphering of colors, especially since other Tables such coding is not used. 

4. Discussion
- I recommend to the authors to rename the titles of subsections 3.1 and 3.2: change 3.1. General on 3.1 General Discussion; change 3.2. More specific on 3.2 Discussions of the Results of Working Hypotheses.
- As far as I understand in subsections 3.2 and 3.3, the authors discuss results concerning working hypotheses. I recommend to the authors to combine subsections 3.3 and 3.2 into one section, taking into account the presence of section 5. Conclusions, there is no need to have so many of them. In addition, by the end of the article, the reader will have long forgotten what was written above. I recommend duplicating these hypotheses, for example: "Hypothesis 1: UHD signal detection using conductivity. Next, give the combined information from subsections 3.2 and 3.3". I think that this approach will facilitate the perception of information for readers and the article itself will be more structured.

5. Other comments
- In the introduction, starting from paragraph 7, it is necessary to remove unnecessary indentation between the paragraphs. Make a red line. Paragraph 7 - lengthen the width of the text. 
- Often in the text there are large paragraph indents between the captions of the figures and the main text. I pay attention to the authors on this and also recommend to the authors to edit the design in accordance with the requirements of the template.

Author Response

Dear Reviewer 1!

Please, see the uploaded file with our answers to your remarks.

Kind regards,

authors

Reviewer 2 Report (New Reviewer)

The paper is devoted for physicochemical study of the molecular signal transfer of ultrahigh diluted antibodies to interferon gamma. The topic is generally interesting, however the paper contain unexplained places (below) and need major revisions.

11) The aim of the paper should be clearly formulated.

22) Figures 1-16 should be more commented and discussed.

33) Experimental procedure should be clearly described.

44) Conclusions should be rewritten in more informative way.

55) English need minor revisions.

EEnglish need minor revisions.

Author Response

Dear Reviewer 2!

Please, see the uploaded file with our answers to your remarks.

Kind regards,

authors

Reviewer 3 Report (New Reviewer)

This paper describes a study about the ultra high dilution (UHD) of substances and its effect on some physico-chemical properties of the solutions. The question about these type of studies is, if there is any change in the observed properties it must be scientifically explained. Indeed, many experimental factors, besides the dilution, will provoque variations in the solution properties, and these variations must be explained and not simply correlated with the dilution. Indeed, many of the observed variations in these type of studies result from uncontrolled experimental factors - and they must be identified. 

Author Response

Dear Reviewer 3!

Please, see the uploaded file with our answers to your remarks.

Kind regards,

authors

Round 2

Reviewer 1 Report (Previous Reviewer 2)

Thanks to the authors for the work done, I recommend this study for publication.

Author Response

Thanks!

Reviewer 2 Report (New Reviewer)

Authors make proper corrections and I suggest publish the paper as it is.

Author Response

Thanks!

Reviewer 3 Report (New Reviewer)

This paper shows a revised version of a manuscript. The main  problem of the raw paper remains, and it is related with the used of very simple measuring techniques, like conductivity and pH, to measured some potential variations that resulted from UHD. Besides quite simple, some of the measurements are being made in highly diluted solutions where the the experimental measurements are highly imprecise and drifts are usually obtained. In order to support the conclusion direct experimental evidence must be obtained. 

Author Response

Please see the attachment (upload).

Round 3

Reviewer 3 Report (New Reviewer)

I cannot observe any further supporting information in the revised paper.

Author Response

NIL

This manuscript is a resubmission of an earlier submission. The following is a list of the peer review reports and author responses from that submission.

Round 1

Reviewer 1 Report

n/a

Reviewer 2 Report

1. Type of the Paper
- At the beginning, I recommend to the authors to specify the type of article.

2. Introduction
- The introduction should be shorter. Thanks to the authors for trying to " break it down for," however, they overdid it. The information in the introduction should be clear and localized. Right now, the information is very unfocused. By the end of the introduction, you forget what was written in the beginning. It shouldn't be this way. I recommend a major revision of the introduction.
- Remove information about your future research, it is superfluous.
- Determine the purpose of your research. There are so many of them now. I recommend that the authors write one overall goal, to achieve which it was necessary to solve the following tasks. And then depending on the solution of these problems to build further presentation of the material. 

3. Results 
- There is no reference in the text to Table 1. Perhaps instead of Table 5, it should be Table 1. I suggest to the authors look into this and make the appropriate changes.
- The first time Table 3 appears in subsection 2.3, however, there is no reference in the text to this Table. Please. make the appropriate changes. 
- All figures and tables are presented incorrectly. This journal has a template which should be used to format the Manuscript. I recommend to the authors to check the Template once again and re-design the figures and tables.
- Subsection 3.3. should be redone. It is inconvenient to go back to the beginning and look for hypothesis number 5 in the introduction. In that case, I recommend to the authors to add a new subsection, for example, 3.3.1 and describe it. 
- Subsection 3.4. should be delete.

4. Materials and Methods
- The methodological part needs to be changed. I recommend that it be reduced to the minimum necessary information. 
- Subsection 4.1. should be delete.

5. Conclusions
- The conclusions need to be redone. It should clearly reflect the solution to the postal problems. 

Minor editing of English language required.

Reviewer 3 Report

The authors of the manuscript "Physicochemical study of the molecular signal transfer of ultra-high diluted antibodies to interferon gamma" study a highly controversial problem of physical effects of ultra high aqueous dilutions of solutes. These effects have been believed to underlie some of homeopathy effects in medical treatments, although there are no convincing proofs of differences in homeopathy treatments from those of placebo administrations. I think the ultra high dilution medical effects should be considered largely as the matter of belief, and in this case they would be nearly as efficient as other non-conventional medications.

From the physical viewpoint, the authors have presented little evidence of the ultra high dilution effects on the solutions. In the boxplots presented in Figures 2-13, the various highly diluted solutions (in fact, chemically pure water) are compared in the conductivity, pH, oxidative-reductive potential, and UV spectrum.  The specifications of the sensors used in the study indicate that the results obtained are within the ranges of accuracy of these sensors (±10 µS/cm for conductivity, ±20 mV for ORP, ±0.2 pH -https://www.vernier.com/product , and 0.005 for the spectrophotometer optical density - https://www.mn-net.com/spectrophotometer-nanocolor-uv/vis-ii-190-1100-nm-919600.1?c=3781). Because the authors do not specify the measurement sequence, it can be thought of as consecutive rather than random. In this case, the measurements can represent, for example, the drift of the instrument indications within the accuracy ranges rather than any meaningful accurate data. The meaningful data could have been obtained only on some random basis of measurements decorrelated in time from the signal drift. Otherwise, I would recommend using more precise instruments for being able to register such subtle differences in the physical parameters associated with the dilution as those presented in the manuscript.

It is no wonder that the most sensitive parameter towards dilution is electrical conductivity. The multiple dilutions (unless performed in clean rooms) drastically increase the probability of capturing ionic impurities (from air) that will increase the conducivity.

The authors do not show the history of the distilled water used for dilutions in terms of the measured physical quantities. It is well known that the equilibration of water with ambient gases such as CO2 or O2 occurs on very slow time scales so that the unequal degrees of inequilibrium may affect the parameters of the diluted solutions in time. In these conditions, the more diluted solutions are expected to be more equilibrated with the ambirnt gases.

Due to the above-mentioned subtle effects of dilution, the specification of as many experimental details as possible is needed for the effects to be reproduced by other workers. For example, the authors should indicate the model of the shaker, its amplitude (and frequency), the exact glassware and volumes used in shaking, etc.

There are also some questions related to the presentation on the material.

93,202: What is electrical tension?

The DOI for the reference: Yinnon, T. Aqueous Solutions and Other Polar Liquids Perturbed by Serial Dilutions and Vigorous Shaking: Analyses of Their UV Spectra. 2018. https://doi.org/10.14294/2018.5 dos not lead to any publication

203: Dissociation into ions

In Tables 1-3 there are no values marked in bold. Does this mean that no significant differences were found between the samples?

The English of the manuscript is comprehensible enough.